# Constrained Cubature Particle Filter for Vehicle Navigation

**DOI:** 10.3390/s24041228

**Published:** 2024-02-15

**Authors:** Li Xue, Yongmin Zhong, Yulan Han

**Affiliations:** 1School of Electronic and Electrical Engineering, Ningxia University, Yinchuan 750021, China; xueli20221@nxu.edu.cn (L.X.); han_yl@nxu.edu.cn (Y.H.); 2School of Engineering, RMIT University, Bundoora, VIC 3082, Australia

**Keywords:** vehicle navigation, constraints, nonlinear filtering, particle filter, constrained cubature particle filter

## Abstract

In vehicle navigation, it is quite common that the dynamic system is subject to various constraints, which increases the difficulty in nonlinear filtering. To address this issue, this paper presents a new constrained cubature particle filter (CCPF) for vehicle navigation. Firstly, state constraints are incorporated in the importance sampling process of the traditional cubature particle filter to enhance the accuracy of the importance density function. Subsequently, the Euclidean distance is employed to optimize the resampling process by adjusting particle weights to avoid particle degradation. Further, the convergence of the proposed CCPF is also rigorously proved, showing that the posterior probability function is converged when the particle number *N* → ∞. Our experimental results and the results of a comparative analysis regarding GNSS/DR (Global Navigation Satellite System/Dead Reckoning)-integrated vehicle navigation demonstrate that the proposed CCPF can effectively estimate system state under constrained conditions, leading to higher estimation accuracy than the traditional particle filter and cubature particle filter.

## 1. Introduction

As the “eye” of a vehicle, the navigation system provides navigation information for vehicle maneuvers to reach their target. Since navigation systems commonly involve nonlinear structural characteristics and dynamics, nonlinear filtering is an important means for navigation computations [1,2,3]. In practice, vehicle navigation systems are subject to constraints and non-Gaussian uncertainties, making nonlinear filter even more challenging [4,5,6].

Currently, the extended Kalman filter (EKF), polynomial filter, unscented Kalman filter (UKF), and cubature Kalman filter (CKF) are commonly used for nonlinear state estimation. The extended Kalman filter involves a system linearization error and requires the calculation of the Jacobian matrix [7,8]. The polynomial filter is sensitive to outliers and requires a large amount of computing for large-scale signal processing [9]. The UKF and CKF select typical sampling points according to prior state estimation to apply the unscented transformation [10,11] and cubature rules [12,13] to obtain sample points, respectively. Subsequently, the obtained sampling points are weighted to acquire system state estimations. Compared to the UKF, the CKF is superior in both estimation accuracy and stability for high-dimensional systems, since the Gaussian-weighted integrals are calculated through third-degree spherical-radial cubature [12,13,14]. In general, these improved nonlinear Kalman filters are subject to the condition that system noises are Gaussian. However, in practical applications (such as multi-sensor integrated systems for vehicle navigation), non-Gaussian noises are commonly involved in nonlinear systems, and thus the direct use of these improved Kalman filters will lead to divergent solutions.

The particle filter (PF) is a typical method used to handle nonlinear systems with non-Gaussian noises [15,16,17]. Instead of integral calculation, PF uses sample mean to implement the Bayesian estimation under nonlinear dynamics. When the particle number is sufficiently large, the posterior probability density function of particles will be sufficiently accurate to guarantee the accuracy of the state mean and variance. However, the particle filter suffers from particle degradation, and there can be difficulty in selecting an appropriate importance density function for importance sampling [18,19,20]. Research efforts have been dedicated to improving the PF. Pitt et al. used auxiliary samples to adjust the resampling order to improve the PF’s performance [21]. D. Liu et al. studied an adaptive partial resampling method to prevent particle degradation [22]. T.H. Liu et al. proposed an adaptive processing scheme to lower particle weights based on a genetic algorithm [23]. However, these improvements cannot guarantee an optimal resultant particle distribution.

The cubature rules provide an effective way to improve sampling accuracy, and thus have been used in the PF to obtain the importance density function, leading to the cubature particle filter (CPF). Xia et al. studied a CPF in the context of system state estimation, where the second-order resistor–capacitor equivalent circuit model was used to verify the filter accuracy [24]. Shi et al. proposed a robust CPF by introducing Huber’s M-estimation theory to handle poor observation data [25]. Feng et al. studied a CPF based on the artificial bee colony for target tracking via underwater wireless sensor networks [26]. Zhang et al. combined a CPF with truncation adaptation to generate recommendation distributions for strong nonlinear systems [27]. Liu et al. investigated a hybrid CPF for stable-state estimation in harsh charging or discharging schedules [28]. Xing et al. constructed an adaptive CPF to estimate navigation parameters for high-dimensional nonlinear vehicle systems [29]. In general, the existing CPFs still suffer from the problem of particle degradation in the PF.

Further, the existing CPFs do not consider constraints. In practical applications (such as multi-sensor integrated systems for vehicle navigation), it is quite common that a nonlinear system is subject to various conditions. Considering these conditions as state constraints in the CPF filter process can effectively improve the estimation accuracy. However, since the existence of state constraints changes the probability structure of a dynamic system, which increases the difficulty of filtering estimation, the related research is still limited. Gao et al. proposed a constrained unscented particle filter for SINS/GNSS/ADS-integrated airship navigation in the presence of wind field disturbances [30]. Seifzadeh et al. studied a constrained particle filter based on soft data to improve filtering performance for target tracking [31]. In complex and highly dynamic environments, Xu et al. presented a particle filter based on spatial–temporal constraints for cooperative target tracking [32]. Zhang et al. presented a constrained multiple model based on a PF for target tracking [33]. However, the above studies are not based on the advanced cubature rules. They also focus on particular applications with specific constraints rather than on general constraint problems.

This paper presents a novel constrained cubature particle filter (CCPF) to improve the CPF’s performance for vehicle navigation. This method incorporates constraints in the cubature transformation to improve the CPF’s importance sampling, resulting in an improved importance density distribution. It also improves the CPF resampling process by using the Euclidean distance of the measurement residual to adjust particle weights to ensure the diversity of particles, thus avoiding particle degradation. The convergence of the proposed CCPF has been rigorously proved. Experiments and a comparative analysis with the traditional PF and CPF were conducted to evaluate the CCPF’s efficiency for GNSS/DR-integrated vehicle navigation.

## 2. Cubature Particle Filter

Consider the nonlinear dynamic system
(1)$$\begin{array}{l} x_{k}=f(x_{k-1})+v_{k}\\ y_{k}=h(x_{k})+n_{k} \end{array}$$
where $x_{k}\in R^{n}$ is the *n*-dimensional system state vector at time point k, $y_{k}\in R^{n}$ is the *m*-dimensional system measurement vector, $v_{k}\in R^{n}$ is the process noise with covariance R, $n_{k}\in R^{m}$ is the measurement noise with covariance Q, and f(⋅) and h(⋅) represent the nonlinear system and measurement functions. Obviously, the system described by (1) does not involve the assumption of Gaussian noises.

Like the unscented transform, CPF uses the third-order spherical radial cubature rules to approximate the nonlinear system functions. According to the prior mean and covariance of the system state, CPF selects the cubature points via the cubature rules and further approximates the state mean and covariance by the weighting sum of the cubature points [34,35,36]. Consider the dynamic system described by (1), the CPF procedure is given as below [37,38]:

Step 1: Initialization

Set the initial state estimate together with its covariance and weights as follows:(2)$$\begin{array}{l} {\hat{x}}_{0}^{j}=E[x_{0}^{j}]\\ P_{0}^{j}=E[(x_{0}^{j}-{\hat{x}}_{0}^{j})(x_{0}^{j}-{\hat{x}}_{0}^{j}){]}^{T}\\ w_{0}^{j}=\frac{1}{N} \end{array}\;(j=1,2,\cdots ,N)$$
where N is the number of particles, ${\hat{x}}_{0}^{j}∼p(x_{0})$ denotes the initial state estimate of the jth particle, and $p(x_{0})$ denotes the initial state distribution.

For k=1,2,⋯,M, execute Steps 2–3.

Step 2: Importance sampling

For j=1,2,⋯,N, execute Steps (a)–(c).

(a) Calculate the cubature points as follows:
(3)$$x_{i,k-1}^{j}=S_{k-1}^{j}\varsigma_{i}+{\hat{x}}_{k-1}^{j}$$
(4)$$P_{k-1}^{j}=S_{k-1}^{j}{\left(S_{k-1}^{j}\right)}^{T}$$
where i=1,2,⋯,2n, $P_{k-1}^{j}$ is the state covariance at k−1, $S_{k-1}^{j}$ is the lower triangular matrix obtained via the Cholesky decomposition of $P_{k-1}^{j}$, $x_{i,k-1}^{j}$ is the *i*th cubature point of the *j*th particle at k−1, ${\hat{x}}_{k-1}^{j}$ is the state estimation, and $\varsigma_{i}$ is defined as follows:
(5)$$\varsigma_{i}=\{\begin{array}{cc} \sqrt{n}I_{i} & i=1,2,\cdots ,n\\ -\sqrt{n}I_{i-n} & i=n+1,n+2,\cdots ,2n \end{array}$$
where $I_{i}$ denotes the *i*th column vector of the n×n identity matrix.

(b) Time update

Calculate the state prediction covariance and one-step measurement prediction via the cubature points
(6)$$x_{i,k|k-1}^{j}=f(x_{i,k-1}^{j})$$
(7)$${\hat{x}}_{i,k|k-1}^{j}=\frac{1}{2n}\sum_{i=1}^{2n}x_{i,k|k-1}^{j}$$
(8)$$P_{k|k-1}^{j}=\frac{1}{2n}\sum_{i=1}^{2n}\left[\left(x_{i,k|k-1}^{j}-{\hat{x}}_{i,k|k-1}^{j}\right){\left(x_{i,k|k-1}^{j}-{\hat{x}}_{i,k|k-1}^{j}\right)}^{T}\right]+R$$
(9)$$y_{k|k-1}^{j}=h({\hat{x}}_{k|k-1}^{j})$$
(10)$${\hat{y}}_{k|k-1}^{j}=\frac{1}{2n}\sum_{i=1}^{2n}y_{k|k-1}^{j}$$
where ${\hat{x}}_{k|k-1}^{j}$, ${\hat{y}}_{k|k-1}^{j}$ and $P_{k|k-1}^{j}$, denote the predicted state mean, measurement mean, and measurement covariance of the *j*th cubature point, respectively.

(c) Measurement update

Calculate the cubature point auto-correlation covariance and cross-correlation covariance
(11)$$P_{y_{k}y_{k}}^{j}=\frac{1}{2n}\sum_{i=1}^{2n}\left[\left(y_{i,k|k-1}^{j}-{\hat{y}}_{k|k-1}^{j}\right){\left(y_{i,k|k-1}^{j}-{\hat{y}}_{k|k-1}^{j}\right)}^{T}\right]+Q\;P_{x_{k}y_{k}}^{j}=\frac{1}{2n}\sum_{i=1}^{2n}\left[\left(x_{i,k|k-1}^{j}-{\hat{x}}_{k|k-1}^{j}\right){\left(y_{i,k|k-1}^{j}-{\hat{y}}_{k|k-1}^{j}\right)}^{T}\right]$$

Calculate the cubature point estimates
(12)$${\hat{x}}_{k}^{j}={\hat{x}}_{k|k-1}^{i}+K_{k}^{j}(y_{k}-{\hat{y}}_{k|k-1}^{j})\;{\hat{P}}_{k}^{j}=P_{k|k-1}^{j}-K_{k}^{j}P_{y_{k}y_{k}}^{j}{\left(K_{k}^{j}\right)}^{T}\;K_{k}^{j}=P_{x_{k}y_{k}}^{j}{\left(P_{y_{k}y_{k}}^{j}\right)}^{-1}$$
where $K_{k}^{j}$ denotes the filter gain at *k* of the *j*th particle, and ${\hat{x}}_{k}^{j}$ and ${\hat{P}}_{k}^{j}$ denote the estimate and its covariance of the *j*th cubature point after cubature transformation.

The particle importance density function is calculated using $\left({\hat{x}}_{k}^{j},{\hat{P}}_{k}^{j}\right)$, and the particles are sampled by ${\bar{x}}_{k}^{j}∼q(x_{k}^{j}|x_{0:k-1}^{j},y_{1:k})=N({\hat{x}}_{k}^{j},{\hat{P}}_{k}^{j})$, ${\bar{x}}_{0:k}^{j}≜(x_{0:k-1}^{j},{\bar{x}}_{k}^{j})$, and ${\hat{P}}_{0:k}^{j}≜(P_{0:k-1}^{j},{\hat{P}}_{k}^{j})$.

Calculate the particle weights and normalize them as follows:(13)$$w_{k}^{j}=w_{k-1}^{j}\frac{p(y_{k}|{\bar{x}}_{k}^{j})p({\bar{x}}_{k}^{j}|x_{k-1}^{j})}{q({\bar{x}}_{k}^{j}|x_{k-1}^{j},y_{k})}$$
(14)$${\tilde{w}}_{k}^{i}=w_{k}^{i}/\sum_{i=1}^{n}w_{k}^{i}$$

Step 3: Resampling

According to the approximately distributed $p{\bar{x}}_{0:k}^{j}|y_{1:k}$ (j=1,2,⋯N), generate N new particles by duplicating the particles of high weights and further assigning the same weight $\frac{1}{N}$ to them.

Calculate the state estimate and its covariance
(15)$${\hat{x}}_{k}=\sum_{j=1}^{N}{\tilde{w}}_{k}^{j}{\bar{x}}_{k}^{j}\;P_{k}=\sum_{j=1}^{N}{\tilde{w}}_{k}^{j}[({\bar{x}}_{k}^{j}-x_{k}){\left({\bar{x}}_{k}^{j}-x_{k}\right)}^{T}]$$

As mentioned previously, the existence of state constraints changes the probability structure of a dynamic system. In the importance sampling step, if the system state is subject to constraints, (7) will be biased. The bias in (7) will be propagated through (8)–(14), and eventually, the state estimate from (15) will be biased. Therefore, it is necessary to incorporate the state changes caused by constraints in the importance sampling step. In addition, in the resampling process, the duplication of the particles with high weights from (13) will lose the particle diversity. Since the useful particle samples are reduced, the filtering solution will deteriorate. A straightforward solution is to adjust the weights of useful particles to increase their contributions to the state estimation. This paper addresses the above two issues in CPF, leading to a new CCPF for state estimation under constraints.

## 3. Constrained Cubature Particle Filter

The proposed CCPF combines Euclidean distance-based resampling and constraints to improve the CPF’s performance. In the importance sampling process, the proposed CCPF improves the CPF importance sampling density function with state constraints. In the resampling process, the proposed CCPF optimizes the CPF weights by adopting the Euclidean distance to adjust the particle weights to ensure the diversity of particles, preventing particle degradation while enabling the easy acquisition of an importance density function which is close to the true density function.

### 3.1. Importance Sampling

In the importance sampling process of CCPF, we perform constrained projection calculations on the cubature point estimates obtained by (12). Suppose the system described by (1) is subject to the following constraint:(16)$$Dx_{k}\leq d$$
where D represents the constrained matrix, and d represents the constrained vector [39,40].

The estimation problem of the constrained state can be transformed into the following optimization problem:(17)$$\min J({\tilde{x}}_{k}^{j})={\left({\tilde{x}}_{k}^{j}-{\hat{x}}_{k}^{j}\right)}^{T}W({\tilde{x}}_{k}^{j}-{\hat{x}}_{k}^{j})$$
(18)$$D{\tilde{x}}_{k}^{j}=d_{k}^{j}$$
where ${\tilde{x}}_{k}^{j}$ represents the state estimation under the constraint, W represents an arbitrary symmetric positive definite matrix for constructing the Lagrange function, and ${\hat{x}}_{k}^{j}$ represents the state estimation without the constraint.

Using the Lagrange multiplier to solve (17) and (18), we can obtain the following:(19)$$J({\tilde{x}}_{k}^{j},\lambda )={\left({\tilde{x}}_{k}^{j}-{\hat{x}}_{k}^{j}\right)}^{T}W({\tilde{x}}_{k}^{j}-{\hat{x}}_{k}^{j})+2\lambda^{T}(D{\tilde{x}}_{k}^{j}-d_{k}^{j})$$
where λ represents a vector for constructing the Lagrange function.

Finding the partial derivative of (19), we obtain
(20)$${\tilde{x}}_{k}^{j}={\hat{x}}_{k}^{j}-W^{-1}D^{T}{\left(DW^{-1}D^{T}\right)}^{-1}(D{\hat{x}}_{k}^{j}-d_{k}^{j})$$

Using (20), the cubature point estimate described by (12) becomes
(21)$${\hat{x}}_{k}^{j}={\hat{x}}_{k|k-1}^{i}+K_{k}^{j}(y_{k}-{\hat{y}}_{k|k-1}^{j})\;{\hat{P}}_{k}^{j}=P_{k|k-1}^{j}-K_{k}^{j}P_{y_{k}y_{k}}^{j}{\left(K_{k}^{j}\right)}^{T}\;K_{k}^{j}=P_{x_{k}y_{k}}^{j}{(}^{-1}P_{x_{k}y_{k}}^{j})\;{\tilde{x}}_{k}^{j}={\hat{x}}_{k}^{j}-W^{-1}D^{T}{\left(DW^{-1}D^{T}\right)}^{-1}(D{\hat{x}}_{k}^{j}-d_{k}^{j})$$

From (21), we can obtain the importance sampling density function under constraints, i.e., ${\bar{x}}_{k}^{j}∼N({\tilde{x}}_{k}^{j},{\hat{P}}_{k}^{*j})$, where ${\hat{P}}_{k}^{*j}$ denotes the covariance under constraints.

### 3.2. Resampling

Calculate the Euclidean distance of the measurement residual. Using (13), record the maximum weights $w_{k}^{j_{\max}}$ and minimum weights $w_{k}^{j_{\min}}$
(22)$$L_{\max}={\left(r_{k}^{j_{\min}}-r_{k}^{j_{\max}}\right)}^{T}\cdot (r_{k}^{j_{\min}}-r_{k}^{j_{\max}})$$
(23)$$L^{j}={\left(r_{k}^{j}-r_{k}^{j_{\max}}\right)}^{T}\cdot (r_{k}^{j}-r_{k}^{j_{\max}})$$
where $L_{\max}$ and $L^{j}$ denote the Euclidean distances with the subscripts “$j_{\max}$” and “$j_{\min}$”, representing the index numbers of the particles with the maximum and minimum weights, and r denotes the measurement residual, i.e.,
(24)$$r_{k}^{j_{\max}}=y_{k}-h({\bar{x}}_{k}^{j_{\max}})\;r_{k}^{j_{\min}}=y_{k}-h({\bar{x}}_{k}^{j_{\min}})$$

Accordingly, the weights are calculated as
(25)$$w_{k}^{j^{*}}=w_{k}^{j}+(\frac{w_{k}^{j_{\max}}}{N})\cdot \sin(\frac{L^{j}}{L_{\max}}\cdot \frac{\pi }{2})\cdot \beta$$
where *β* ≥ 0 is a coefficient related to measurement characteristics [41]. The larger β is, the larger the adjustment to the weight will be. When β=0, there will be no adjustment to the weight. The CCPF procedure is shown in Figure 1.

### 3.3. Convergence Analysis

In this section, we will study the convergence of the proposed CCPF when the sample size N is sufficiently large. Since the posterior density of the system state corresponds to the empirical measure of particle samples, we conduct a convergence analysis based on an empirical measure which is defined by a probability measure.

Similar to the PF, assuming that the transfer kernel function ϕ satisfies the Feller process and the state likelihood function g is a continuous bounded positive definite function (i.e., ‖g‖<∞), for k≥0, when N→∞, we can have [42]
(26)$$\underset{N\to \infty }{\lim}\pi_{k|k}^{N}=\underset{N\to \infty }{\lim}\phi_{1:k}^{N}(\eta^{N})=\phi_{1:k}(\eta )=\pi_{k|k}$$
where $\pi_{k|k}$ denotes the ideal distribution of the system state, η represents the initial distribution, and $\pi_{k|k}^{N}$ is the posterior probability density, which is expressed as
(27)$$\pi_{k|k}^{N}(dx_{k})=\frac{1}{N}\sum_{i=1}^{N}\delta_{x_{k}^{i}}(dx_{k})$$
where δ denotes the Dirichlet function.

The time and measurement updates of CCPF can be represented as follows [43]:(28)πk|k−1=∫ℝnxπk−1|k−1(dxk−1)ϕ(dxk|xk−1)
(29)πk|k(dxk)=g(yk|xk)πk|k−1(dxk)∫ℝnxg(yk|xk)πk|k−1(dxk)

Equations (28) and (29) can be rewritten as
(30)(πk|k−1,φ)=(πk−1|k−1,ϕφ)
(31)(πk|k,φ)=(πk|k−1,g)−1(πk|k−1,φg)
where φ⊂B(ℝnx) can be any bounded function, and (πk|k−1,g)>0.

As shown in Section 3.1, the CCPF involves three key steps: the initialization, importance sampling, and resampling. Therefore, the convergence analysis of CCPF will be applied to each step. Denote the posterior probability density in the importance sampling process using $\pi_{k|k}^{N}$ and that in the resampling process using ${\tilde{\pi }}_{k|k}^{N}$.

**Theorem** **1.***For any bounded function* φ⊂B(ℝnx)*;* *if* ‖g‖ *is bounded, then* 
(32)E[((πk−1|k−1N,φ)−(πk−1|k−1,φ))2]≤ck−1|k−1N‖φ‖2N

**Proof.** Using (31), we obtain
(33)$$\begin{array}{ll} \left(\pi_{k-1|k-1}^{N},\varphi )-(\pi_{k-1|k-1},\varphi \right) & =\frac{\left(\pi_{k-1|k-2}^{N},g\varphi \right)}{\left({\tilde{\pi }}_{k-1|k-2}^{N},g\right)}-\frac{\left(\pi_{k-1|k-2},g\varphi \right)}{\left(\pi_{k-1|k-2},g\right)}\\ & =\frac{\left(\pi_{k-1|k-1}^{N},g\varphi \right)}{\left(\pi_{k-1|k-1}^{N},g\right)}-\frac{\left(\pi_{k-1|k-2},g\varphi \right)}{\left(\pi_{k-1|k-2}^{N},g\right)}+\frac{\left(\pi_{k-1|k-2},g\varphi \right)}{\left(\pi_{k-1|k-2}^{N},g\right)}--\frac{\left(\pi_{k-1|k-2},g\varphi \right)}{\left(\pi_{k-1|k-2}^{N},g\right)}\\ & =\frac{\left(\pi_{k-1|k-2}^{N},g\varphi \right)-\left(\pi_{k-1|k-2},g\varphi \right)}{\left(\pi_{k-1|k-2}^{N},g\right)}\\ & +\frac{\left(\pi_{k-1|k-2},g\varphi \right)(\left(\pi_{k-1|k-2},g\varphi \right)-\left(\pi_{k-1|k-2}^{N},g\right))}{\left(\pi_{k-1|k-2}^{N},g\right)\left(\pi_{k-1|k-2},g\right)} \end{array}$$According to the Minkowski inequality, we have
(34)$$\begin{array}{l} E[(\frac{\left(\pi_{k-1|k-2}^{N},g\varphi )-(\pi_{k-1|k-2},g\varphi \right)}{\left(\pi_{k-1|k-2}^{N},g\right)}+\frac{\left(\pi_{k-1|k-2},g\varphi )((\pi_{k-1|k-2},g\varphi )-(\pi_{k-1|k-2}^{N},g)\right)}{\left(\pi_{k-1|k-2}^{N},g)(\pi_{k-1|k-2},g\right)}{)}^{2}{]}^{\frac{1}{2}}\\ \leq E[(\frac{\left(\pi_{k-1|k-2}^{N},g\varphi )-(\pi_{k-1|k-2},g\varphi \right)}{\left(\pi_{k-1|k-2}^{N},g\right)}{)}^{2}{]}^{\frac{1}{2}}+E[(\frac{\left(\pi_{k-1|k-2}^{\;},g\varphi )((\pi_{k-1|k-2}^{\;},g\varphi )-(\pi_{k-1|k-2}^{N},g)\right)}{\left(\pi_{k-1|k-2}^{N},g)(\pi_{k-1|k-2}^{\;},g\right)}{)}^{2}{]}^{\frac{1}{2}}\\ \leq \frac{E[\left((\pi_{k-1|k-2}^{N},g\varphi )-(\pi_{k-1|k-2}^{\;},g\varphi \right){)}^{2}{]}^{\frac{1}{2}}}{\left(\pi_{k-1|k-2}^{N},g\right)}+\frac{(\pi_{k-1|k-2}^{\;},g\varphi )E[\left((\pi_{k-1|k-2}^{\;},g\varphi )-(\pi_{k-1|k-2}^{N},g\varphi \right){)}^{2}{]}^{\frac{1}{2}}}{\left(\pi_{k-1|k-2}^{N},g)(\pi_{k-1|k-2}^{\;},g\right)}\\ \leq \frac{\sqrt{c_{k-1|k-2}^{\;}}\left\|g\varphi \right\|}{({\tilde{\pi }}_{k-1|k-2}^{N},g)\sqrt{N}}+\frac{E[\left((\pi_{k-1|k-2}^{N},g\varphi )-(\pi_{k-1|k-2}^{\;},g\varphi \right){)}^{2}{]}^{\frac{1}{2}}}{\left(\pi_{k-1|k-2}^{N},g\right)} \end{array}$$Since ‖g‖<∞, we have
(35)$$\begin{array}{l} \frac{\sqrt{c_{k-1|k-2}^{\;}}\left\|g\varphi \right\|}{({\tilde{\pi }}_{k-1|k-2}^{N},g)\sqrt{N}}+\frac{E[\left((\pi_{k-1|k-2}^{N},g\varphi )-(\pi_{k-1|k-2}^{\;},g\varphi \right){)}^{2}{]}^{\frac{1}{2}}}{\left(\pi_{k-1|k-2}^{N},g\right)}\\ \leq \frac{\sqrt{c_{k-1|k-2}^{\;}}\left\|g\varphi \right\|}{(\pi_{k-1|k-2}^{N},g)\sqrt{N}}+\frac{\sqrt{c_{k-1|k-2}^{\;}}\left\|g\varphi \right\|}{(\pi_{k-1|k-2}^{N},g)\sqrt{N}}\\ \leq \frac{2\sqrt{c_{k-1|k-2}^{\;}}\left\|g\varphi \right\|}{(\pi_{k-1|k-2}^{N},g)\sqrt{N}}=\sqrt{c_{k-1|k-1}^{N}}\frac{\left\|\varphi \right\|}{\sqrt{N}} \end{array}$$
where $\sqrt{c_{k-1|k-1}^{N}}=\frac{2\sqrt{c_{k-1|k-2}^{\;}}\left\|g\right\|}{\left(\pi_{k-1|k-2}^{N},g\right)}$.Equation (32) follows by substituting (35) into (34) and further applying the square operation. Thus, the proof of (32) is completed.Theorem 1 shows that the initialization process of the CCPF is converged when N→∞. □

**Theorem** **2.***If* ‖ϕφ‖≤‖φ‖ *holds, the importance sampling process of the CCPF yields*(36)$$E[\left((\pi_{k|k-1}^{N},\varphi )-(\pi_{k|k-1}^{\;},\varphi \right){)}^{2}]\leq c_{k|k-1}^{N}\frac{{\left\|\varphi \right\|}^{2}}{N}$$

**Proof.** Using (30), we can obtain
(37)$$\begin{array}{l} E[((\pi_{k|k-1}^{N},\varphi )-(\pi_{k|k-1}^{\;},\varphi ){)}^{2}{]}^{\frac{1}{2}}\\ =E[((\pi_{k|k-1}^{N},\varphi )-(\pi_{k-1|k-1}^{N},\phi \varphi )+(\pi_{k-1|k-1}^{N},\phi \varphi )-(\pi_{k|k-1}^{\;},\varphi ){)}^{2}{]}^{\frac{1}{2}}\\ \leq E[((\pi_{k|k-1}^{N},\varphi )-(\pi_{k-1|k-1}^{N},\phi \varphi ){)}^{2}{]}^{\frac{1}{2}}+E[(\pi_{k-1|k-1}^{N},\phi \varphi )-(\pi_{k-1|k-1}^{\;},\phi \varphi ){)}^{2}{]}^{\frac{1}{2}}\\ =E[((\pi_{k-1|k-1}^{N},\phi \varphi^{2})-(\pi_{k-1|k-1}^{N},\phi \varphi ){)}^{2}{]}^{\frac{1}{2}}+E[(\pi_{k-1|k-1}^{N},\phi \varphi )-(\pi_{k-1|k-1}^{\;},\phi \varphi ){)}^{2}{]}^{\frac{1}{2}} \end{array}$$According to Theorem 1 and letting ${\left(1+\sqrt{c_{k-1|k-1}^{\;}}\right)}^{2}=c_{k|k-1}^{N}$, and when ‖ϕφ‖≤‖φ‖ holds, we can have
(38)$$\begin{array}{l} E[((\pi_{k-1|k-1}^{N},\phi \varphi^{2})-(\pi_{k-1|k-1}^{N},\phi \varphi ){)}^{2}{]}^{\frac{1}{2}}+E[(\pi_{k-1|k-1}^{N},\phi \varphi )-(\pi_{k-1|k-1},\phi \varphi ){)}^{2}{]}^{\frac{1}{2}}\\ \leq \frac{\;}{\sqrt{N}}+\sqrt{c_{k-1|k-1}^{\;}}\frac{\left\|\varphi \right\|}{\sqrt{N}}\\ =(1+\sqrt{c_{k-1|k-1}^{\;}})\frac{\left\|\varphi \right\|}{\sqrt{N}} \end{array}$$Equation (36) follows by substituting (38) into (37) and further applying the square operation. Thus, the proof of Theorem 2 is completed.Theorem 2 shows that the importance sampling process of the CCPF is converged when N→∞. □

The resampling process of the CCPF involves the adjustment of the particle weights. In theory, the prior distribution should be close to the importance sampling distribution. Then, using (13), we have $w_{k}^{j}\propto g$. From (25) and $\left|\left(\frac{w_{k}^{j_{\max}}}{N})\cdot \sin(\frac{L^{j}}{L_{\max}}\cdot \frac{\pi }{2}\right)\right|<1$, it is readily known that the particle weights are bounded. By normalization, we can readily have the particle weights within [0, 1]. Without the loss of generality and for the concise description purpose, we write g→Fg, where *F* is a linear transformation, and Fg is bounded. Thus, we can derive the following theorem.

**Theorem** **3.**
*For the resampling process of the CCPF, we have*

(39)
$$E[(({\tilde{\pi }}_{k|k}^{N},\varphi )-(\pi_{k|k},\varphi ){)}^{2}]\leq c_{k|k}^{N}\frac{{\left\|\varphi \right\|}^{2}}{N}$$



**Proof.** Using (31) and g→Fg, we can obtain the following:(40)$$\begin{array}{l} E[(({\tilde{\pi }}_{k|k}^{N},\varphi )-(\pi_{k|k}^{\;},\varphi ){)}^{2}{]}^{\frac{1}{2}}\\ =E[(({\tilde{\pi }}_{k|k}^{N},\varphi )-({\tilde{\pi }}_{k|k}^{N},\varphi )+({\tilde{\pi }}_{k|k}^{N},\varphi )-(\pi_{k|k}^{\;},\varphi ){)}^{2}{]}^{\frac{1}{2}}\\ =E[(\frac{\left({\tilde{\pi }}_{k|k-1}^{N},(Fg)\varphi \right)}{\left({\tilde{\pi }}_{k|k-1}^{N},Fg\right)}-\frac{\left(\pi_{k|k-1}^{\;},(Fg)\varphi \right)}{\left({\tilde{\pi }}_{k|k-1}^{N},Fg\right)}+\frac{\left(\pi_{k|k-1},(Fg)\varphi \right)}{\left({\tilde{\pi }}_{k|k-1}^{N},Fg\right)}-\frac{\left(\pi_{k|k-1},(Fg)\varphi \right)}{\left(\pi_{k|k-1}^{\;},Fg\right)}{)}^{2}{]}^{\frac{1}{2}}\\ =E[((\frac{\left({\tilde{\pi }}_{k|k-1}^{N},(Fg)\varphi \right)}{\left({\tilde{\pi }}_{k|k-1}^{N},Fg\right)}-\frac{\left(\pi_{k|k-1},(Fg)\varphi \right)}{\left({\tilde{\pi }}_{k|k-1}^{N},Fg\right)})+(\frac{\left(\pi_{k|k-1},(Fg)\varphi \right)}{\left({\tilde{\pi }}_{k|k-1}^{N},Fg\right)}-\frac{\left(\pi_{k|k-1},(Fg)\varphi \right)}{\left(\pi_{k|k-1},Fg\right)}){)}^{2}{]}^{\frac{1}{2}}\\ \leq E[(\frac{\left({\tilde{\pi }}_{k|k-1}^{N},(Fg)\varphi \right)}{\left({\tilde{\pi }}_{k|k-1}^{N},Fg\right)}-\frac{\left(\pi_{k|k-1}^{\;},(Fg)\varphi \right)}{\left({\tilde{\pi }}_{k|k-1}^{N},Fg\right)}{)}^{2}{]}^{\frac{1}{2}}+E[(\frac{\left(\pi_{k|k-1}^{\;},(Fg)\varphi \right)}{\left({\tilde{\pi }}_{k|k-1}^{N},Fg\right)}-\frac{\left(\pi_{k|k-1}^{\;},(Fg)\varphi \right)}{\left(\pi_{k|k-1}^{\;},Fg\right)}{)}^{2}{]}^{\frac{1}{2}} \end{array}$$According to Theorem 2 and the letting constant $c_{k|k}^{N}=\frac{2\sqrt{c_{k|k-1}}\left\|Fg\right\|}{\left({\tilde{\pi }}_{k|k-1}^{N},Fg\right)}$, we have
(41)$$\begin{array}{l} E[(\frac{\left({\tilde{\pi }}_{k|k-1}^{N},(Fg)\varphi \right)}{\left({\tilde{\pi }}_{k|k-1}^{N},Fg\right)}-\frac{\left(\pi_{k|k-1}^{\;},(Fg)\varphi \right)}{\left({\tilde{\pi }}_{k|k-1}^{N},Fg\right)}{)}^{2}{]}^{\frac{1}{2}}+E[(\frac{\left(\pi_{k|k-1}^{\;},(Fg)\varphi \right)}{\left({\tilde{\pi }}_{k|k-1}^{N},Fg\right)}-\frac{\left(\pi_{k|k-1}^{\;},(Fg)\varphi \right)}{\left(\pi_{k|k-1}^{\;},Fg\right)}{)}^{2}{]}^{\frac{1}{2}}\\ =E[(\frac{\left({\tilde{\pi }}_{k|k-1}^{N},(Fg)\varphi )-(\pi_{k|k-1}^{\;},(Fg)\varphi \right)}{\left({\tilde{\pi }}_{k|k-1}^{N},Fg\right)}{)}^{2}{]}^{\frac{1}{2}}+E[(\frac{\left(\pi_{k|k-1}^{\;},(Fg)\varphi )((\pi_{k|k-1}^{\;},Fg)-({\tilde{\pi }}_{k|k-1}^{N},Fg\right)}{\left({\tilde{\pi }}_{k|k-1}^{N},Fg)(\pi_{k|k-1}^{\;},Fg\right)}{)}^{2}{]}^{\frac{1}{2}}\\ \leq \frac{\sqrt{c_{k|k-1}^{\;}}\left\|Fg\right\|\cdot \left\|\varphi \right\|}{({\tilde{\pi }}_{k|k-1}^{N},g)\sqrt{N}}+\frac{\left\|\varphi \right\|\cdot E[\left((\pi_{k|k-1}^{\;},Fg)-({\tilde{\pi }}_{k|k-1}^{N},Fg\right){)}^{2}{]}^{\frac{1}{2}}}{\left({\tilde{\pi }}_{k|k-1}^{N},g\right)}\\ \leq \frac{2\sqrt{c_{k|k-1}^{\;}}\left\|Fg\right\|\cdot \left\|\varphi \right\|}{({\tilde{\pi }}_{k|k-1}^{N},Fg)\sqrt{N}}=\sqrt{c_{k|k}^{N}}\frac{\left\|\varphi \right\|}{\sqrt{N}} \end{array}$$Since ‖Fg‖<∞, (39) follows by substituting (41) into (40) and further applying the square operation. Thus, the proof of (39) is completed.Theorem 3 shows that the resampling process of the CCPF is converged when N→∞. □

From Theorems 1–3, it can be inferred that the posterior probability function of CCPF will be converged when N→∞, which indicates the convergence of the proposed CCPF. 

## 4. Experimental Results

### 4.1. GNSS/DR Vehicle Navigation System

Experiments on a GNSS/DR-integrated navigation system of a sports car were conducted to evaluate the performance of the proposed CCPF. The state vector of the GNSS/DR-integrated navigation system is defined as follows:(42)$$x(t)={\left[\begin{array}{cccccccc} p_{E} & v_{E} & a_{E} & p_{N} & v_{N} & a_{N} & \varepsilon & \psi \end{array}\right]}^{T}$$
where $p_{E}$, $v_{E}$, and $a_{E}$ are the position, velocity, and acceleration in east; $p_{N}$, $v_{N}$, and $a_{N}$ are the position, velocity, and acceleration in the north; ε is the gyro drift error; and ψ is the DR calibration coefficient.

The system state equation is as follows [44,45]:(43)$$\dot{x}=\left[\begin{array}{cccccccc} 0 & 1 & 0 & 0 & 0 & 0 & 0 & 0\\ 0 & 0 & 1 & 0 & 0 & 0 & 0 & 0\\ 0 & 0 & -\frac{1}{\tau_{a_{E}}} & 0 & 0 & 0 & 0 & 0\\ 0 & 0 & 0 & 0 & 1 & 0 & 0 & 0\\ 0 & 0 & 0 & 0 & 0 & 1 & 0 & 0\\ 0 & 0 & 0 & 0 & 0 & -\frac{1}{\tau_{a_{N}}} & 0 & 0\\ 0 & 0 & 0 & 0 & 0 & 0 & \frac{1}{\tau_{\varepsilon }} & 0\\ 0 & 0 & 0 & 0 & 0 & 0 & 0 & 0 \end{array}\right]x(t)+u+v$$
where v is the process noise with covariance R; $\tau_{a_{E}}$ and $\tau_{a_{N}}$ are the correlation time constants for the variation rates of the accelerations in the east and north; $\tau_{\varepsilon }$ is the correlation time constant for the first-order Markov process in the gyro drift; and u is the control input, defined as
(44)$$u={\left[\begin{array}{cccccccc} 0 & 0 & \frac{{\bar{a}}_{E}}{\tau_{a_{E}}} & 0 & 0 & \frac{{\bar{a}}_{N}}{\tau_{a_{N}}} & 0 & 0 \end{array}\right]}^{T}$$
where ${\bar{a}}_{E}$ and ${\bar{a}}_{N}$ are the means of the accelerations in the east and north at the current time.

The measurement vector y is defined as
(45)$$y={\left[\begin{array}{cccc} p_{E} & p_{N} & \omega & s \end{array}\right]}^{T}$$
where $p_{E}$ and $p_{N}$ represent the positions in the east and north from the GNSS receiver, ω is the angular rate from the gyroscope, and s is the output distance from the DR.

The system measurement equation is described as follows:(46)y=h(x)+n
where h(⋅) is the nonlinear measurement function, and n is the measurement noise with covariance Q.
(47)$$h\left(x\right)=\{\begin{array}{c} p_{E}\\ p_{N}\\ \frac{v_{N}a_{E}-v_{E}a_{N}}{v_{E}^{2}+v_{N}^{2}}+\varepsilon \\ \psi T\sqrt{v_{E}^{2}+v_{N}^{2}} \end{array}$$
where ε is the first-order Markov process component of the gyro drift error, and T is the sampling period.

According to the road conditions restricting the system state, the car travelling direction was constrained on the road to the east by the angle θ. The constraint equation is expressed as follows:(48)$$Dx=d=0\;D=\left[\begin{array}{cccccccc} -tg\theta & 0 & 0 & 1 & 0 & 0 & 0 & 0\\ 0 & -tg\theta & 0 & 0 & 1 & 0 & 0 & 0 \end{array}\right]$$

### 4.2. Experimental Setup

The experimental setup is shown in Figure 2. The high-precision differential GNSS receiver UT-206 with the positioning accuracy less than 10 cm was used to obtain the actual vehicle position as the reference to calculate the estimation error.

The GNSS data update rate was 1 Hz. The GNSS single-point L1/L2 accuracy was 1.5 m in the vertical direction and 1 m in the horizontal direction. The GNSS receiver accuracy was 0.5 m in the vertical direction and 0.3 m in the horizontal direction. The GNSS velocity accuracy was 0.05 m/s. During the test, at least seven navigation satellite signals were available for the GNSS measurement. The experimental data were collected from the car, which was travelling within a continuous time of 1000 s. After initialization for 1 min, the car moved stably at an average velocity of 40 km/h. The car travelling direction was constrained to the east by θ=65°. The calibration coefficient of DR was ψ=0.5. The drift error of the gyroscope was 0.1°/h, and $\tau_{E}=\tau_{N}=300\;s$. The car’s initial position, velocity, and acceleration were (0 m, 0 m), (5 m/s, 5 m/s), and (0 m/s^2^, 0 m/s^2^). The total mileage was 15 km. $R=\mathrm{diag}[{\begin{array}{cccccccc} {\left(10\;m\right)}^{2} & {\left(1\;m/s\right)}^{2} & {\left(0.1\;m/s^{2}\right)}^{2} & {\left(10\;m\right)}^{2} & {\left(1\;m/s\right)}^{2} & {\left(0.1\;m/s^{2}\right)}^{2} & {\left(0.1^{\circ }/h\right)}^{2} & \left(1\;m\right) \end{array}}^{2}]$, and ***Q*** = *diag*[(20 m)^2^ (20 m)^2^ (4 m/s)^2^ (4 m/s)^2^].

For comparison purposes, experiments were conducted using the PF, CPF, and CCPF to estimate the car’s position and velocity errors. The root mean squared error (RMSE) was used as the metric for accuracy evaluation. The RMSE is defined as follows:(49)$$\mathrm{RMSE}=\sqrt{\frac{1}{F}\sum_{i=1}^{F}({\hat{x}}_{i}-x_{ref})}$$
where *F* is the number of Monte Carlo runs, and $x_{ref}$ is the reference value. For the experimental analysis, 1000 Monte Carlo runs were conducted for the RMSE calculation. The overall RMSE is defined as follows:(50)$${\mathrm{RMSE}}_{overall}^{\;}=\sqrt{{\mathrm{RMSE}}_{E}^{2}+{\mathrm{RMSE}}_{N}^{2}}$$
where ${\mathrm{RMSE}}_{E}$ and ${\mathrm{RMSE}}_{N}$ denote the RMSEs in the east and north.

Figure 3, Figure 4 and Figure 5 show the position errors obtained by the PF, CPF, and CCPF. It can be seen that the position error curves of all the three methods involve large fluctuations within the initial 100 s due to the system initialization. After the initial 100 s, the PF curves of position error still fluctuate greatly, leading to a position error of (−13.1 m, 14.0 m) in the north and a position error of (−14.5 m, 16.1 m) in the east. The CPF improves the PF with the cubature rules for particle sampling, leading to the position errors within (−13.0 m, 11.1 m) the north and (−13.0 m, 12.1 m) east. However, since the CPF still suffers from particle degradation and does not involve constraints in the state estimation, its improvement is limited. In contrast, since the CCPF optimizes the CPF sampling procedures, its position error is (−8.1 m, 9.6 m) in the north and (−7.0 m, 7.1 m) in the east. Table 1 lists the position RMSEs of the PF, CPF, and CCPF. The position RMSEs of the PF, CPF, and CCPF are 4.1719 m, 3.7231 m, and 2.7061 m in the north and 4.7618 m, 3.7504 m, and 2.5168 m in the east. The overall position RMSEs of the PF, CPF, and CCPF are 6.3308 m, 5.2846 m, and 3.6956 m, showing that the accuracy of the CCPF is 41.6% higher than that of the PF and 30.1% higher than that of the CPF.

Figure 6, Figure 7 and Figure 8 illustrate the velocity errors obtained by the PF, CPF, and CCPF. The velocity errors of the three methods have a similar trend to their position errors shown in Figure 3, Figure 4 and Figure 5. The velocity errors of the PF, which are (−0.24 m/s, 0.22 m/s) in the north and (−0.25 m/s, 0.21 m/s) in the east, is the largest. The CPF slightly improves the PF, leading to velocity errors of (−0.11 m/s, 0.16 m/s) in the north and (−0.18 m/s, 0.19 m/s) in the east. In contrast, the velocity errors of the CCPF, which are (−0.14 m/s, 0.15 m/s) (north) and (−0.10 m/s, 0.15 m/s) (east), are the smallest. Table 2 summarizes the velocity RMSEs of the three methods. The velocity RMSEs of the PF, CPF, and CCPF are 0.0912 m/s, 0.0892 m/s, and 0.0769 m/s in the north and 0.1023 m/s, 0.0887 m/s, and 0.0674 m/s in the east. The overall velocity RMSEs of the PF, CPF, and CCPF are 0.1371 m/s, 0.1258 m/s, and 0.1022 m/s, showing that the accuracy of the CCPF is 25.5% higher than that of the PF and 18.8% higher than that of the CPF.

From the above results, it can be concluded that the CCPF can achieve more accurate state estimations under constrained conditions than the CPF and PF for GNSS/DR-integrated navigation. It is also shown that the filtering solution of CCPF is bounded, which further verifies the theoretical convergence analysis of the CCPF (described in Section 3.3).

The above RMSEs, together with the theoretical convergence analysis described in Section 3.3, demonstrate the consistent performance of the proposed CCPF. To further evaluate the filter’s consistency, Figure 9 illustrates quantile–quantile plots of the position residuals derived from the use of the PF, CPF, and CCPF. As shown in Figure 9a, it is obvious that the quantile of the position residuals of the PF does not coincide with its normal quantile. Although the residual quantiles of both the CPF and CCPF are in good agreement with their normal quantiles, as shown in Figure 9b,c, the residual quantile of the CCPF is much closer to its normal quantile compared to that of the CPF. Therefore, it is clear that the proposed CCPF possesses the consistency for state estimation.

## 5. Conclusions

Nonlinear systems with constraints and non-Gaussian uncertainties are commonly encountered in vehicle navigation. To address this issue, this paper proposes a new CCPF to estimate system state parameters for vehicle navigation. It enhances the CPF importance sampling process using constraints to improve the importance density distribution. Subsequently, it improves the CPF resampling process by using the Euclidean distance of the measurement residual to adjust particle weights to ensure the diversity of particles, thus decreasing particle degradation. Theories were also established to prove the convergence of the proposed CCPF. The experimental results and the results derived from our comparative analysis of GNSS/DR-integrated vehicle navigation demonstrate that the proposed CCPF can effectively estimate system state parameters in constrained environments, resulting in it having enhanced accuracy compared to the PF and CPF.

Our future research work will focus on the improvement of the proposed CCPF. The proposed CCPF will be combined with advanced artificial intelligence technologies and genetic algorithms to achieve nonlinear state estimation in the environments of more complex constraints.

## Figures and Tables

**Figure 1 sensors-24-01228-f001:**
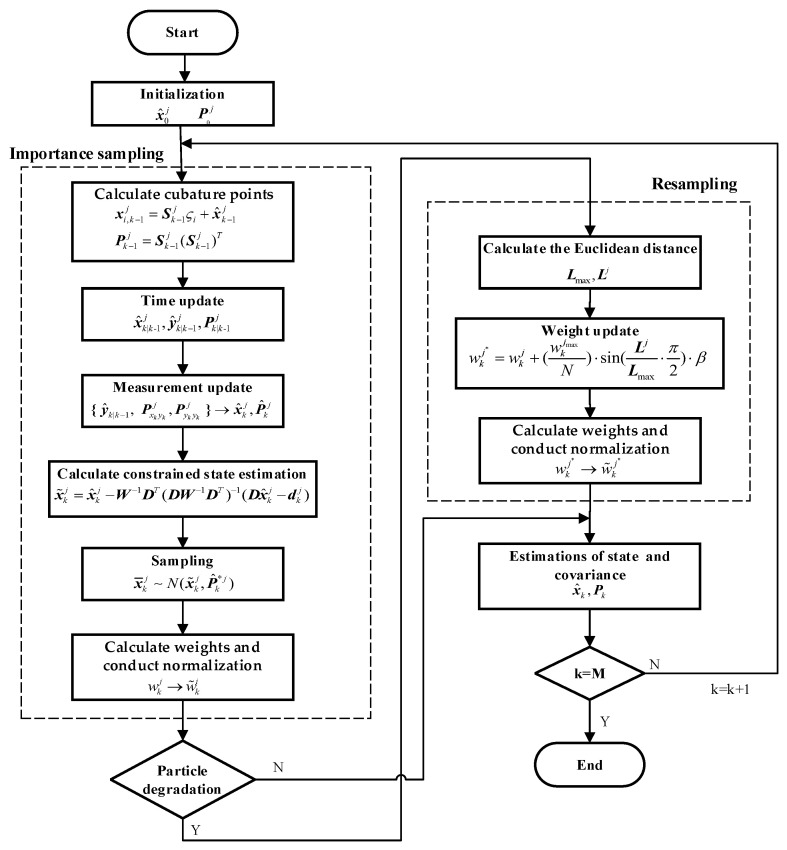
The CCPF procedure.

**Figure 2 sensors-24-01228-f002:**
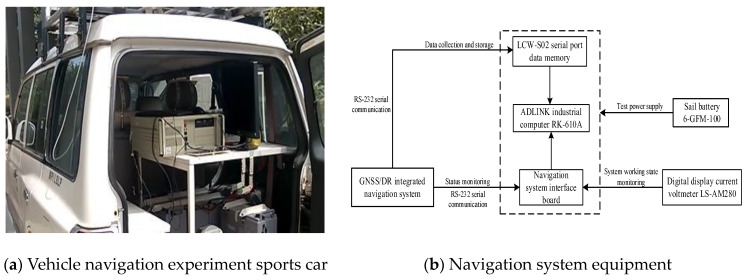
Experimental setup.

**Figure 3 sensors-24-01228-f003:**
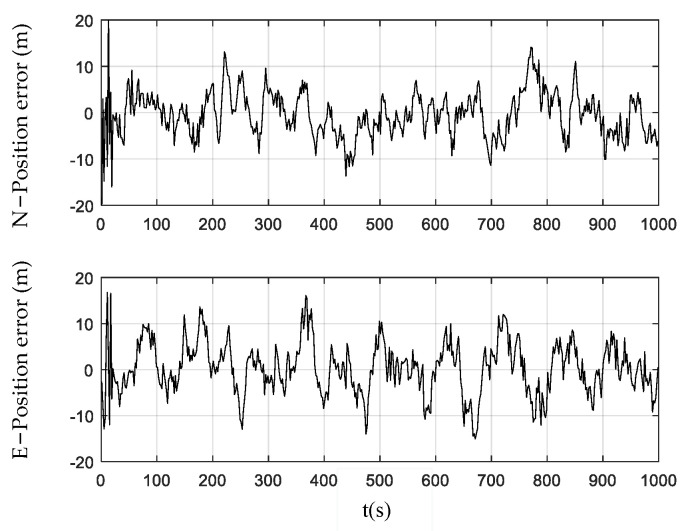
Position error obtained using the PF.

**Figure 4 sensors-24-01228-f004:**
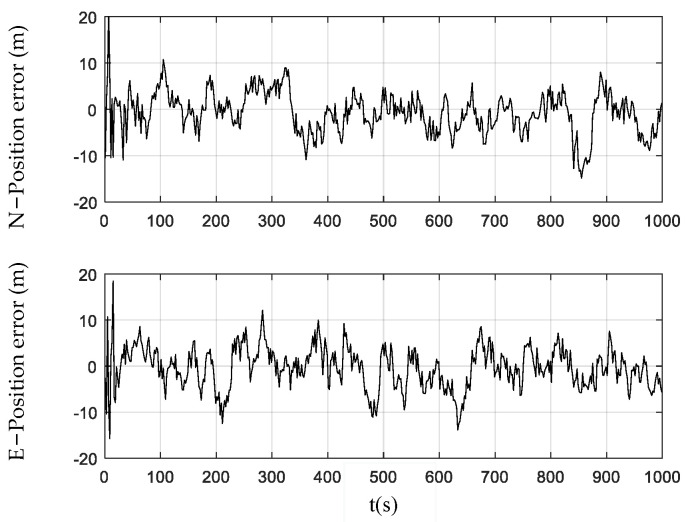
Position error obtained using the CPF.

**Figure 5 sensors-24-01228-f005:**
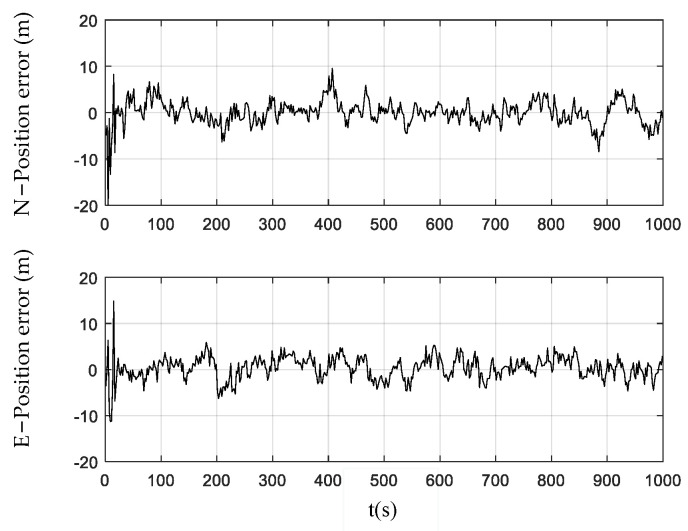
Position error obtained using the CCPF.

**Figure 6 sensors-24-01228-f006:**
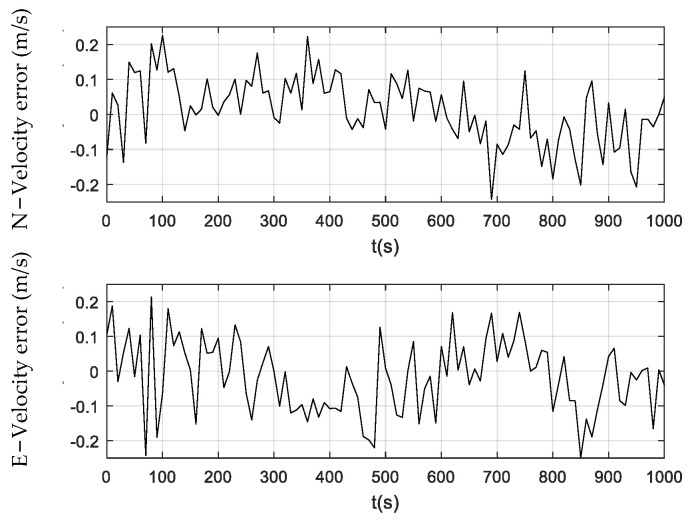
Velocity error obtained using the PF.

**Figure 7 sensors-24-01228-f007:**
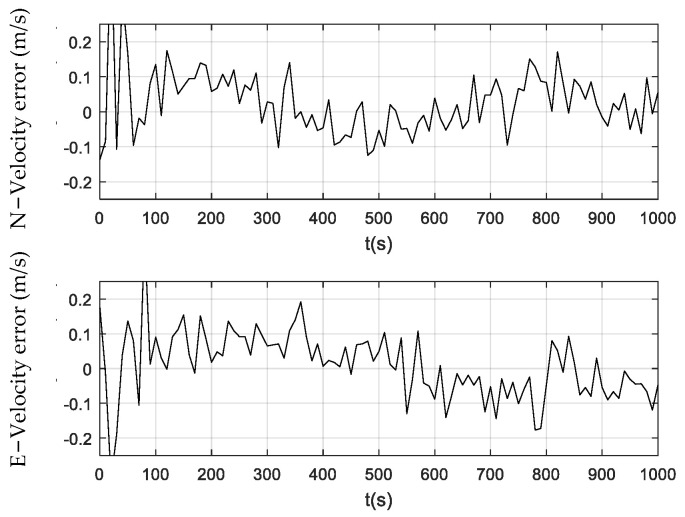
Velocity error obtained using the CPF.

**Figure 8 sensors-24-01228-f008:**
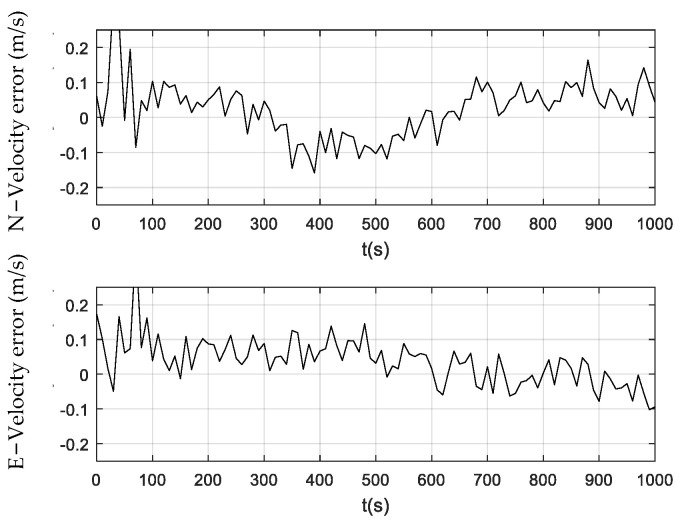
Velocity error obtained using the CCPF.

**Figure 9 sensors-24-01228-f009:**
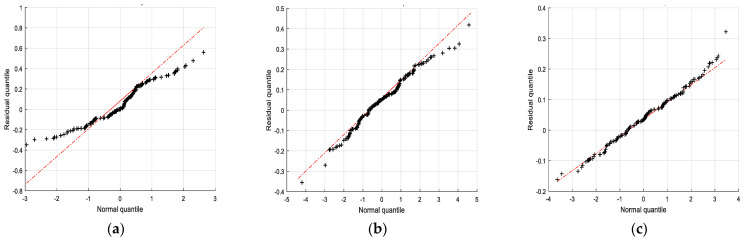
Quantiles of the position residuals derived from the use of (**a**) the PF, (**b**) the CPF, and (**c**) the CCPF. The quantiles of the position residuals are indicated by the black dots and “+”, and the normal quantiles are indicated by the dashed red line.

**Table 1 sensors-24-01228-t001:** Position RMSEs of the PF, CPF, and CCPF.

Filter	RMSE in East (m)	RMSE in North (m)	Error Range (m)	Overall RMSE (m)
PF	4.1719	4.7618	(−13.1, 14.0), (−14.5, 16.1)	6.3308
CPF	3.7231	3.7504	(−13.0, 11.1), (−13.0, 12.1)	5.2846
CCPF	2.7061	2.5168	(−8.1, 9.6), (−7.0, 7.1)	3.6956

**Table 2 sensors-24-01228-t002:** Velocity RMSEs of the PF, CPF, and CCPF.

Filter	RMSE in East (m/s)	RMSE in North (m/s)	Error Range (m/s)	Overall RMSE (m/s)
PF	0.0912	0.1023	(−0.24, 0.22), (−0.25, 0.21)	0.1371
CPF	0.0892	0.0887	(−0.11, 0.16), (−0.18, 0.19)	0.1258
CCPF	0.0769	0.0674	(−0.14, 0.15), (−0.10, 0.15)	0.1022

## Data Availability

The data used to support the findings of this study are available from the corresponding author upon request.

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
