# Peer review of "Constrained Cubature Particle Filter for Vehicle Navigation"

_sensors, 2024, doi:10.3390/s24041228_

Round 1

Reviewer 1 Report

Comments and Suggestions for Authors

A new constrained cubature particle filter is proposed in the paper for vehicle navigation solution. Experimental and comparison analyses on vehicle navigation with GNSS/DR integration demonstrate that the proposed method can effectively estimate system state under constrained conditions. But there are still some problems :

1. Figure 3-5 has no abscissa variables and units.

2. The description of Figs.2-4 on pages 306 and 307 is incorrect.

3. The English description of some sentences in the text needs to be modified and improved.

Comments on the Quality of English Language

The English description of some sentences in the text needs to be modified and improved.

Reviewer 2 Report

Comments and Suggestions for Authors

This article has made improvements to cubature particle filtering, which is innovative and supported by clear targeted experiments. The article mentions the application of research laboratory in vehicle navigation in the abstract, but there is relatively little research content on vehicle navigation filtering in the introduction and main text, and there are not many related literature in the references. This makes the research content of this article not closely related to vehicle navigation. It is recommended to add relevant content on vehicle navigation research in the introduction and text, and also to include more relevant literature in the references, so that the research content of this article is not closely related to vehicle navigation

Reviewer 3 Report

Comments and Suggestions for Authors

The paper considers a modification of the cubature filter, the designing of which takes into account additional a priori information about the estimated vector. The accuracy of the algorithms is analyzed using the example of the problem of integrated data processing from a GNSS and a dead reckoning system.

The paper is written sloppily and contains many inaccuracies both in the formulas and in the text.

1.     Lines 32, 58 It is necessary to refer to key works on the discussed algorithms (UKF, CKF, CPKF), as well as second-order filters or polynomial filters, which are often used to solve nonlinear problems. See for example:

Gustafsson F. Hendeby G. Some Relations Between Extended and Unscented Kalman Filters. Signal Processing, IEEE Transactions on, 2012. Vol. 60. №2. P. 545-555.

Stepanov, O.A., Litvinenko, Y.A., Vasiliev, V.A. et al. Polynomial Filtering Algorithm Applied to Navigation Data Processing under Quadratic Nonlinearities in System and Measurement Equations. Part 1. Description and Comparison with Kalman Type Algorithms. Gyroscopy Navig. 12, 205–223 (2021).

2.     The part “Cubature particle filter” comes after the introduction. A section with a statement of the problem is required.

3.     Line 116 It is required to clarify that a bank of j cubature filters is used.

4.     From formula (4) it follows that ζ is a n dimensional vector. From formula 5 it follows that ζ is a matrix.

5.     Formulas (6) and further: you need to check all the formulas and correct typos and inaccuracies. For example, the equations (6) for predicting of the state vector and Eq. (9) for  predicting of the measurements are inconsistent with equations(1).

6.     Line 135. Formula (15) assumes that all particles and their corresponding weights have already been calculated. Apparently the phrase “Repeat Steps 2-3 to process all particle samples” should have been written earlier?

7.     Line 170. Figure 1. From the diagram it follows that in the case when the weights do not degenerate, the algorithm starts working again and an infinite loop is closed.

8.     Line 230 et seq. The paper must use uniform notation. For example, in part 2 (line 99) Q is the variance of measurement noise .In (43) Q is the system noise variance matrix.

9.     Line 270. It is not clear whether the experiment was carried out in simulation mode or using the real data?

10.  Line 266. An important characteristic of algorithms used to solve nonlinear problems of processing navigation information, in addition to accuracy, is also consistency (correspondence of the calculated accuracy characteristic generated in the algorithm to its real values). When comparing algorithms, the issue of consistency also needs to be considered.

Reviewer 4 Report

Comments and Suggestions for Authors

This manuscript introduces an innovative approach to enhance the accuracy and robustness of vehicle navigation by integrating state constraints into the Cubature Particle Filter (CPF). This integration addresses the challenge of particle degradation, improving overall state estimation. The proposed method refines the importance sampling process within CPF by incorporating state constraints, thereby enhancing the accuracy of the importance density function.   Additionally, it utilizes Euclidean distance during the resampling process to adjust particle weights, optimizing diversity and preventing particle degradation. This paper has been significantly improved after the first round of revisions, but there are still some problems in the manuscript:

1.     Novelty and comprehensiveness of referencesThe references in this paper predominantly draw from sources dated 2022 and earlier, with limited inclusion of classic positioning schemes and strategies. To address this, it is imperative to update the references accordingly and reorganize the introduction section of the paper. Such as:

(1)   Zhao, X., Min, H., Xu, Z., & Wang, W. (2019). An ISVD and SFFSD-based vehicle ego-positioning method and its application on indoor parking guidance. Transportation Research Part C: Emerging Technologies, 108, 29-48. DOI: 10.1016/j.trc.2019.09.001

(2)   Yuan, Y., Li, F., Chen, J., Wang, Y., & Liu, K. (2024). An improved Kalman filter algorithm for tightly GNSS/INS integrated navigation system. Mathematical Biosciences and Engineering, 21(1), 963-983. DOI: 10.3934/mbe.2024040

2.     Insufficient empirical evidence: While the text outlines the proposed method and its convergence analysis, it lacks empirical evidence or experimental results to substantiate the presented claims.  Without such validation, evaluating the effectiveness or practicality of the proposed approach becomes challenging.

3.     Absence of comparison with existing methods: The text omits any mention of how the proposed method fares against existing approaches or methods in the field.  A comprehensive evaluation and comparison with established techniques would enable readers to grasp the novelty and advantages of the proposed method.

Comments on the Quality of English Language

This manuscript introduces an innovative approach to enhance the accuracy and robustness of vehicle navigation by integrating state constraints into the Cubature Particle Filter (CPF). This integration addresses the challenge of particle degradation, improving overall state estimation. The proposed method refines the importance sampling process within CPF by incorporating state constraints, thereby enhancing the accuracy of the importance density function.   Additionally, it utilizes Euclidean distance during the resampling process to adjust particle weights, optimizing diversity and preventing particle degradation. This paper has been significantly improved after the first round of revisions, but there are still some problems in the manuscript:

1.     Novelty and comprehensiveness of referencesThe references in this paper predominantly draw from sources dated 2022 and earlier, with limited inclusion of classic positioning schemes and strategies. To address this, it is imperative to update the references accordingly and reorganize the introduction section of the paper. Such as:

(1)   Zhao, X., Min, H., Xu, Z., & Wang, W. (2019). An ISVD and SFFSD-based vehicle ego-positioning method and its application on indoor parking guidance. Transportation Research Part C: Emerging Technologies, 108, 29-48. DOI: 10.1016/j.trc.2019.09.001

(2)   Yuan, Y., Li, F., Chen, J., Wang, Y., & Liu, K. (2024). An improved Kalman filter algorithm for tightly GNSS/INS integrated navigation system. Mathematical Biosciences and Engineering, 21(1), 963-983. DOI: 10.3934/mbe.2024040

2.     Insufficient empirical evidence: While the text outlines the proposed method and its convergence analysis, it lacks empirical evidence or experimental results to substantiate the presented claims.  Without such validation, evaluating the effectiveness or practicality of the proposed approach becomes challenging.

3.     Absence of comparison with existing methods: The text omits any mention of how the proposed method fares against existing approaches or methods in the field.  A comprehensive evaluation and comparison with established techniques would enable readers to grasp the novelty and advantages of the proposed method.

Reviewer 5 Report

Comments and Suggestions for Authors

In this paper, a constrained cubature particle filter for vehicle navigation is designed and seems to achieve the desired results as seen from experiments. The workload for this article is sufficient. However, I still have some suggestions and questions for the author to consider and this paper should be major revised before making the final decision.

1. Equation (2), what does If stand for. The authors are requested to check the paper for similar problems and correct them.

2. There are obvious grammatical errors in the paper. For example, page 4, ”To obtains N new particle.....”. The writer should scrutinize the grammar of the sentences in this paper and correct any errors.

3. The paper mentions that Kalman filtering is not suitable for state estimation in non-Gaussian noise, so the authors should also set up Kalman filtering in comparative experiments to confirm the superiority of the proposed algorithm.

4. In Fig. 8 the velocity tracking error of the CCPF is 10^(-1) level, is this valuable in practical applications? Also in other studies on vehicle navigation, what is the accuracy of state estimation done by others?

5. According to Fig. 1, is the difference between CCPF and CPF only in “Calculate constrained value?”

Comments on the Quality of English Language

In this paper, a constrained cubature particle filter for vehicle navigation is designed and seems to achieve the desired results as seen from experiments. The workload for this article is sufficient. However, I still have some suggestions and questions for the author to consider and this paper should be major revised before making the final decision.

1. Equation (2), what does If stand for. The authors are requested to check the paper for similar problems and correct them.

2. There are obvious grammatical errors in the paper. For example, page 4, ”To obtains N new particle.....”. The writer should scrutinize the grammar of the sentences in this paper and correct any errors.

3. The paper mentions that Kalman filtering is not suitable for state estimation in non-Gaussian noise, so the authors should also set up Kalman filtering in comparative experiments to confirm the superiority of the proposed algorithm.

4. In Fig. 8 the velocity tracking error of the CCPF is 10^(-1) level, is this valuable in practical applications? Also in other studies on vehicle navigation, what is the accuracy of state estimation done by others?

5. According to Fig. 1, is the difference between CCPF and CPF only in “Calculate constrained value?”

Round 2

Reviewer 3 Report

Comments and Suggestions for Authors

The added part regarding the consistency of the algorithm requires a clearer description. So it remains unclear whether the consistency study was carried out or not.

Reviewer 5 Report

Comments and Suggestions for Authors

Accept in present form

Comments on the Quality of English Language

Accept in present form

Author Response

Dear Reviewer,

This is a positive comment, and thus no modification is needed. We would like to express our great appreciation to you for previous constructive comments, which have contributed to the progress of this paper. Thank you for your recognition of this paper.

Best regards